# Membrane Lipid Replacement with Glycerolphospholipids Slowly Reduces Self-Reported Symptom Severities in Chemically Exposed Gulf War Veterans

Garth L. Nicolson * and Paul C. Breeding

Department of Molecular Pathology, The Institute for Molecular Medicine, Huntington Beach, CA 92647, USA; drpaulcbreeding@yahoo.com
* Correspondence: gnicolson@immed.org; Tel.: +1-949-715-5978

**Abstract:** Background: Chemically exposed veterans of the 1991 Gulf War have few options for treatment of conditions and symptoms related to their chemical exposures. Membrane Lipid Replacement (MLR) with oral membrane glycerolphospholipids is a safe and effective method for slowly removing hydrophobic organic molecules from tissues, while enhancing mitochondrial function and decreasing the severity of certain signs and symptoms associated with multi-symptom illnesses. Methods: A preliminary open-label study utilizing 20 male veterans who were deployed to combat areas, exposed to environmental toxic chemicals and subsequently diagnosed with Gulf War Illnesses (GWI) were utilized. These subjects took 6 g per day oral glycerolphospholipids for 6 months, and the severities of over 100 signs and symptoms were self-reported at various times using illness survey forms. Results: In the sixteen patients that fully complied and completed the study, there were gradual and significant reductions of symptom severities in categories related to fatigue, pain, musculoskeletal, nasopharyngeal, breathing, vision, sleep, balance, and urinary, gastrointestinal and chemical sensitivities. There were no adverse incidents during the study, and the all-natural oral study supplement was extremely well tolerated. Conclusions: MLR with oral glycerolphospholipids appears to be a simple, safe and potentially effective method of slowly reducing the severities of multiple symptoms in chemically exposed veterans.

**Keywords:** chemical exposures; multi-symptom illnesses; Gulf War veterans; Gulf War Illnesses; membrane lipid supplements; detoxification; symptom survey forms

## 1. Introduction

In late 1990 to early 1991, approximately 700,000 armed forces personnel from the USA along with personnel from 30 coalition countries were deployed to the Persian Gulf in support of military operations (Operation Desert Shield/Operation Desert Storm) collectively known as the 1991 Gulf War [1,2]. After returning from the Kuwaiti Theater of Operations (KTO), large numbers of veterans reported persistent, deployment-related health problems when compared to non-deployed personnel [3–5]. The most common signs and symptoms found included: chronic fatigue, arthralgia, myalgia, headaches, gastrointestinal problems, sleeping difficulties, dermatological symptoms, breathing problems, loss of concentration and cognition, depression, muscle spasms, nervousness, blurred vision, anxiety, chest and heart pain, dizziness, nausea, stomach pain, loss of balance, hives, frequent coughing, chemical sensitivities, eye pain, vision problems and photophobia, bleeding gums and other symptoms [4–7]. This multi-symptom chronic illness has been termed Gulf War Illness (GWI) [7,8].

Although explanations for GWI have been advanced, such as stress, infections, chemoprophylactic agents, physical trauma and exposures to environmental conditions, including sand, chemicals and other exposures, conclusions on the causes of GWI remain unproven [5–13]. Many of these veterans may eventually have their diagnoses linked, in

part, to chemical exposures, such as oil spills and fires, smoke from military operations, chemicals on clothing, pesticides, chemoprophylactic agents (pyridostigmine bromide), chemical weapons and other possible exposures [2,6,8,9]. However, in some veterans, the apparent spread of a similar illness to immediate family members in the absence of environmental sources suggests that a minority of veterans were exposed to one or more infection(s) [7,10,11]. In addition, fine sand exposure was rather ubiquitous during the deployment to the KTO, and continued pulmonary exposure to fine sand particles can result in hyperergic lung conditions, and in more severe cases pneumonitis [12]. The presumed exposures to quite different, multiple agents and toxicants have made the successful treatment of GWI quite difficult.

The treatments of GWI patients have depended, at least in some cases, on the types of exposures presumed to be present in individual GWI cases [7]. Treatment approaches have been complicated by multiple exposures (chemical, biological and, in a few cases, radiological as well as stress) encountered in the KTO and the lack of available methods to successfully treat or remove offending substances [7,13]. Here, we used an exceptionally safe approach, Membrane Lipid Replacement (MLR) [14,15], which can slowly and safely remove hydrophobic molecules, and possibly some of the chemicals implicated in GWI [2,13]. Membrane glycerolphospholipids form the lipid matrix of biological membranes and have been extensively studied [15,16]. This removal process is thought to occur by a bulk flow or mass action mechanism where hydrophobic molecules are slowly sequestered into small glycerolphospholipid droplets and eventually removed from cells and tissues and deposited into the gastrointestinal system for elimination [16,17]. In preliminary case reports, the oral use of MLR glycerolphospholipids resulted in dramatic reductions in fatigue, pain and other symptoms in Gulf War veterans [18].

MLR using oral supplements can result in the systemic replacement of damaged cellular membrane glycerolphospholipids with undamaged, unoxidized lipids to ensure the proper function of cellular membranes, including mitochondrial membranes, and potentially facilitate the removal of hydrophobic chemical toxicants [14–16,19]. By combining the glycerolphospholipids with fructooligosaccahrides to protect the phospholipids from disruption, degradation and oxidation in the gut and antioxidants to protect against oxidative damage, MLR supplements have proven to be effective in reducing disease-associated symptom severities and age-associated loss of function. MLR provides organ, tissue and cell membrane support while enhancing mitochondrial function [14–16]. Here, we used MLR in an attempt to sequester and remove cellular hydrophobic toxicants and reduce symptom severities in chemically exposed Gulf War veterans. Case reports had previously suggested the usefulness of MLR in Gulf War veterans in lowering some symptom severities [18,20]. Here, we examined a wide range of signs and symptoms in a patient cohort composed of combat veterans exposed to chemicals and diagnosed with GWI in order to determine whether MLR is an effective approach to reducing morbidity and improving function in these patients.

## 2. Materials and Methods

An open-label, Institutional Review Board-approved, preliminary or pilot clinical study was initiated to study the effects of an all-natural glycerolphospholipid chewable wafer supplement (Patented Energy® with NTFactor Lipids®) on the severities of signs and symptoms of GWI patients. Five supplement wafers providing a total of 6 g of the test supplement NTFactor Lipids® were taken each day. The supplement wafers were provided by Nutritional Therapeutics, Inc. (Hauppuage, NY, USA). This supplement is a patented, proprietary membrane lipid preparation containing exogenous plant polyunsaturated phosphatidylcholine, phosphatidyglyerol, phosphatidylserine, phosphatidylinositol and other membrane phospholipids and fructooligosaccharides to protect the phospholipids from disruption, degradation and oxidation, as well as antioxidants to protect against oxidative damage [19,20]. The participants took the supplement daily dose for 6 months, and their signs and symptoms were self-reported at various times (0, 0.25, 1, 3 and 6 months)

using a patient symptom survey form (Supplemental Figure S1). Symptom severities in the survey form were scored numerically based on a linear scale from 0 to 10 or lowest (0) to highest (10) severity of symptoms. The symptom severity form contained approximately 120 signs and symptoms that were merged into 22 symptom categories to more easily assess the effects of the supplement (Table 1).

**Table 1.** Symptom categories based on the Signs and Symptoms Survey Form.

| Category | Signs or Symptoms from the Trial Symptom Survey Form |
| --- | --- |
| Heart | palpitations, chest pain, skipped beats, racing pulse, chest pressure |
| Nasopharyngeal | congestion, nasal discharge, sinus pain, sore throat, sneezing, coughing, throat clearing |
| Breathing | unable to breathe deeply, wheezing, shortness of breath |
| Neurological | depression, loss of interest, suicidal thoughts, mood swings, memory loss, irritability, headaches, concentration |
| Sleep | insomnia, nightmares, unrefreshed sleep |
| Fatigue | chronic fatigue, malaise, tiredness |
| Pain | widespread pain, skeletomuscular pain, neck pain, back pain, headache |
| Infections | fever, night sweats, swollen glands, frequent colds, mouth sores |
| Hair/Scalp | hair loss, discoloration, scalp itch |
| Skin | rashes, reddening, itching, peeling, color change, slow wound healing, growths on skin, fungus, sores |
| Gastrointestinal | stomach pain, cramps, flatus, diarrhea, bloating, blood in stool, nausea, vomiting, excessive thirst, loss of interest in food |
| Urinary | bladder control, frequent urination, blood in urine |
| Oral cavity | loose teeth, bleeding gums, abscesses, increased salivation, dry mouth, hoarseness, coated tongue, lip sores |
| Vision | blurred vision, double vision, loss of night vision, loss of acuity. light sensitivity, floaters, twitching, dry eyes, itchy eyes, watery eyes |
| Balance | poor balance, vertigo, steadiness |
| Senses | tinnitus, hearing loss, loss of smell, loss of taste, cold sensitivity |
| Audial | stuttering, difficulty finding words, numbness of lips, drooling |
| Joints | joint pain, loss of mobility, ache |
| Muscles | muscle pain, burning, spasms, ache, cramps, loss of strength, swelling |
| Allergy | increased allergic sensitivities, increased sensitivities to biologicals |
| Genital | genital pain, itching, swelling, impotence |
| Chemical Sensitivity | sensitivities to fumes, exhaust, smoke, fuel |

*2.1. Patients*

Twenty GWI patients were recruited using social media and asked to provide information on their service units, deployments and their health before, during and after their deployment to the KTO in late 1990 to early 1991. Of this group, 14 were former U.S. Army soldiers, 5 were former U.S. Marines and one was a U.S. Navy sailor. During deployment, all of the participants received multiple military vaccines, including the anthrax vaccine. All of the participants were deployed to KTO combat zones where they remained until after the war, and all signed a Patient Informed Consent document. Based on their self-reported signs and symptoms before, during and after the Gulf War and before starting the open label study, all of the participants had multi-symptom chronic illnesses with moderate to severe signs and symptoms. These signs and symptoms were consistent with the two most commonly used criteria for diagnosis of GWI using the chronic multi-symptom illness standards developed by the U.S. Government and the State of Kansas [3,21]. Inclusion criteria in the present study included at least 3 out of 6 symptom categories: fatigue, pain, neurological, skin, gastrointestinal, and respiratory symptoms [21]. Exclusion criteria included diagnosis of a serious medical condition not usually associated with deployment to the KTO or a psychiatric condition that could account for some symptoms or interfere with accurate symptom scoring. Additionally, the participants who did not comply completely with the reporting requirements or did not take all of the required supplement wafers were excluded from the study results. The average age in years of participants that fully completed the study was 53.1 ± 3.2.

### 2.2. Study Design

Male subjects with GWI signs and symptoms who signed an Informed Consent document and agreed to participate in the study were sent the test supplement and the Quantitative Signs and Symptoms Survey Forms, which also contained information on environmental exposures and clinical diagnoses after the Gulf War (Supplemental Figure S1). Each participant was instructed to complete the Symptom Survey Form on Day 0 before taking the test supplement in the morning (3 wafers) and evening (2 wafers) for a total of 6 g per day of the test supplement. The reporting occurred on Day 0, and at the end of Day 7 (Week 1), Month 1, Month 3 and Month 6 and was recorded within 24 h of the prescribed dates. During and after the study, the completed symptom survey forms were returned to the Principal Investigator in pre-paid mailers provided to each participant. Patients were advised to not to change any of their daily medications, diet or routine during the study.

### 2.3. Statistics

Data were analyzed by analysis of variance (ANOVA), with significance defined as $p < 0.05$ or better (Intellectus Statistics$^{TM}$). Further data analysis was performed with the Tukey test, with significance defined as $p < 0.05$. The standardized alpha (Cronbach's alpha) was used to confirm reliability and internal consistency of the data [22].

### 3. Results

### 3.1. Subjects in the Study

Sixteen out of twenty subjects (all male) were fully compliant and completed the study. The main reason for participants not completing the study was non-compliance with trial instructions. Either the subject did not return the signed Informed Consent document, did not take all of the study supplements or they failed to send back all of the fully completed symptom survey forms.

Subjects in the study had received a variety of individual diagnoses, but all had a diagnosis of GWI, a multi-symptom chronic illness (Figure 1). All of the participants in the study received physical examinations before deployment and were considered in good health before being deployed to the KTO.

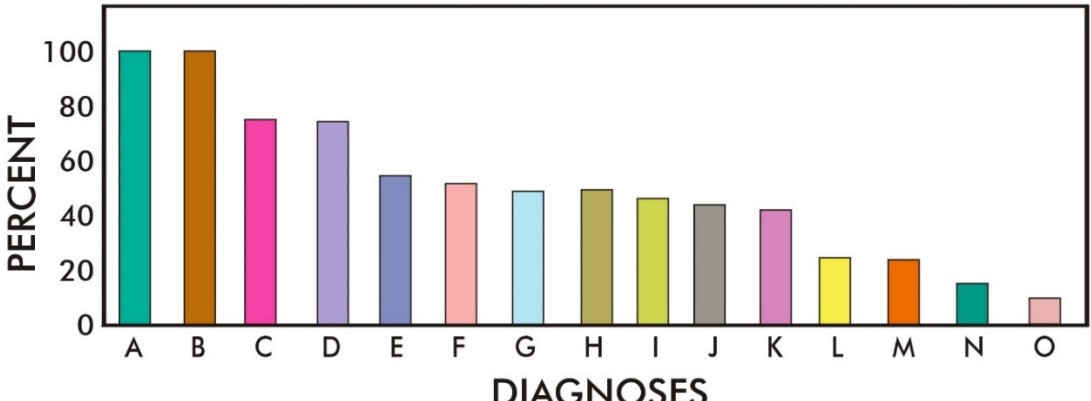

**Figure 1.** Diagnoses and/or symptoms of study participants reported after the 1991 Gulf War. All participants had multiple symptoms or diagnoses at various times before entering the trial. Percent of participants with: A: Gulf War illnesses, B: chronic fatigue syndrome, C: multiple chemical sensitivity syndrome, D: weight gain/loss, E: irritable bowel syndrome, F: fibromyalgia syndrome, G: allergies, H: skin rashes, I: depression, J: joint pain, K: hypertension, L: reactive airway symptoms, M: sleep disturbances, N: skin lesions, O: contact dermatitis.

### 3.2. Toxic Exposures

During the Gulf War study, participants were exposed to a variety of environmental conditions and toxicants (Figure 2). For example, all of the participants reported exposures

to oil well fires and burn pit smoke, fuels, and crude or refined oils (A–C in Figure 2). In addition, all of the participants took oral chemoprophylactic agents (pyridostigmine bromide) during the conflict (D in Figure 2). Most participants reported that they were also exposed to pesticides (E in Figure 2), but less so to insects (G in Figure 2). Some trial participants also reported exposure to raw sewage (F in Figure 2) and dead bodies (H in Figure 2). A few participants may have been exposed to chemical warfare agents (I in Figure 2), and there was a very low incidence of reported exposure to herbicides (J in Figure 2).

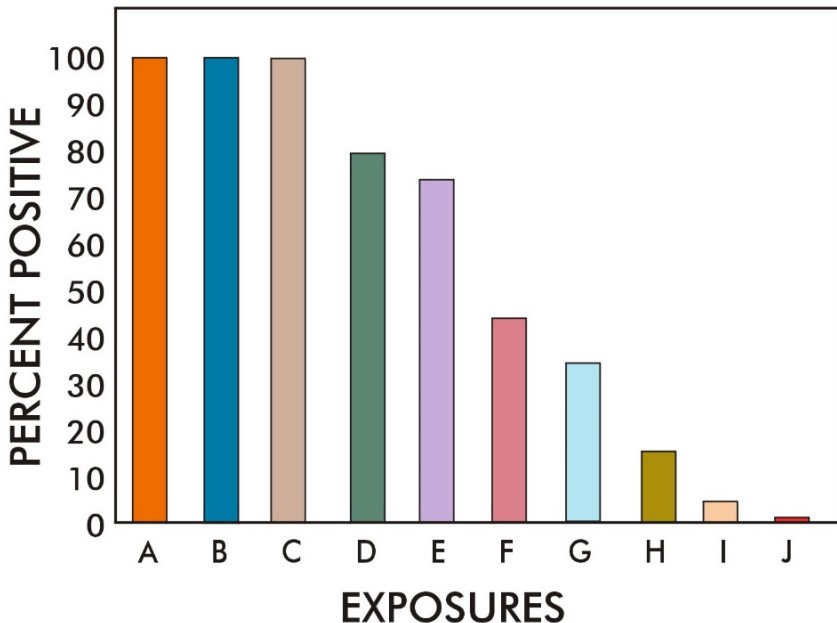

**Figure 2.** Trial participant-reported exposures during the 1991 Gulf War. Percent of trial participants exposed to: A: direct fuel and/or oil, B: smoke from burning oil wells, C: smoke from burn pits, D: ingestion of pyridostigmine bromide, E: exposure to pesticides, F: exposure to raw sewage, G: exposure to insects, H: direct contact with dead bodies, I: presumed exposure to chemical warfare agents, and J: exposure to herbicides.

*3.3. Symptom Severities during the Study*

At the beginning of the study, subjects reported a variety of signs and symptom severities that were different in each patient (Figure 3). In Figure 3, the self-reported symptoms and their severities were grouped into symptom categories (Table 1), and the reported category severities and their change over time were different for each patient. The changes in two study participants' self-reported symptoms with time during the study are shown in Figure 3 (green and blue symbols). These two participants can be compared with the mean ± standard deviations of symptom severities of the entire group (red symbols).

Although there were differences in the symptom category severities and responses to the test supplement in individual subjects, the group mean data indicated that there were gradual and significant responses to the test supplement. Specifically, there were significant reductions in symptom category severities related to fatigue, pain, musculoskeletal, nasopharyngeal, breathing, vision, sleep, balance, gastrointestinal, chemical sensitivities and other symptom categories (Figure 3). As expected, these reductions were gradual and significant ($p < 0.01$ to $p < 0.001$) with time (6 months) in various symptom categories, such as fatigue (Figure 3F), sleep disfunction (Figure 3E), gastrointestinal symptoms (Figure 3J), pain (Figure 3V) and other symptoms. In the present study, we also considered symptoms such as self-reported changes in chemical sensitivities (Figure 3T).

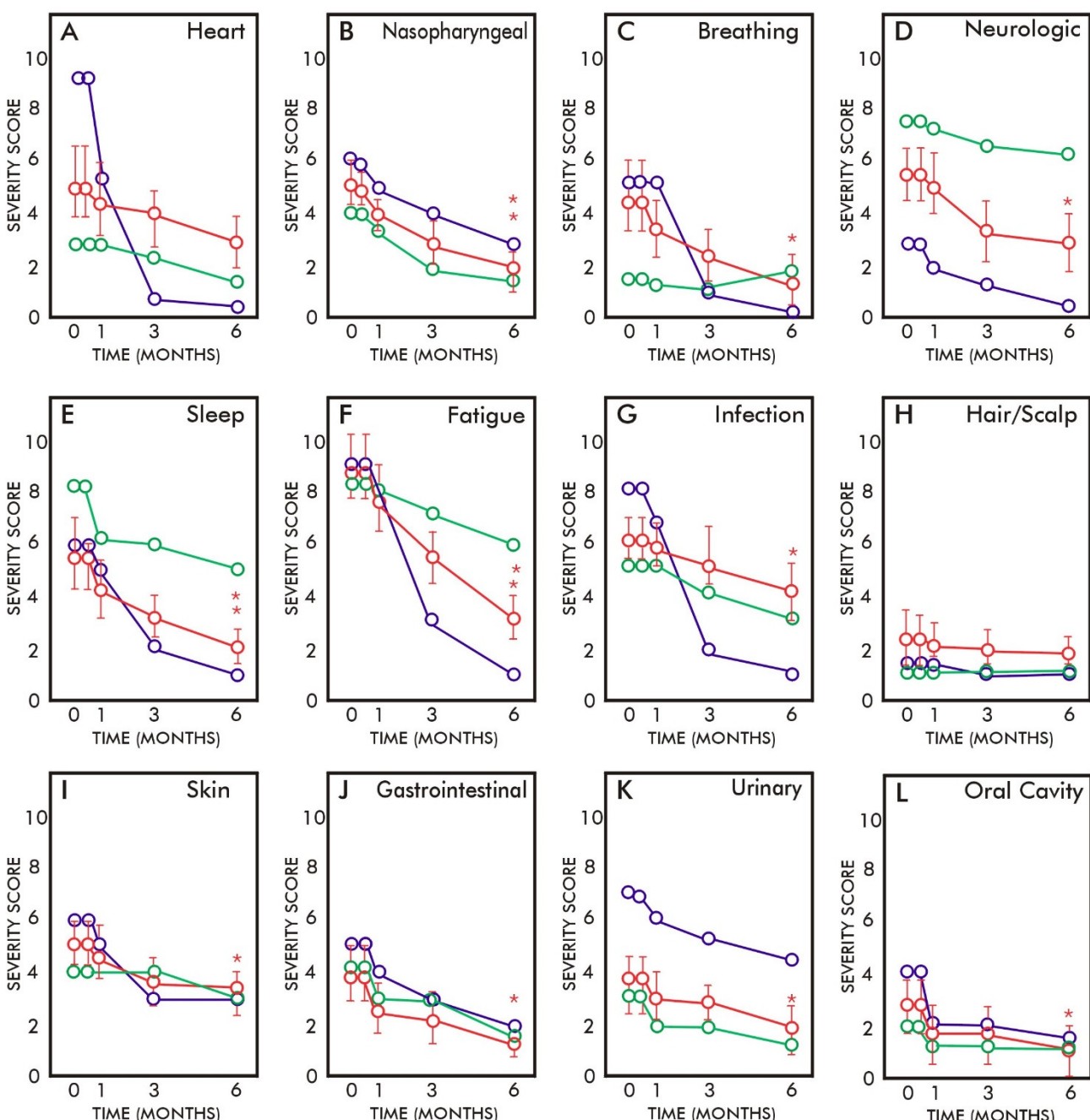

**Figure 3.** *Cont.*

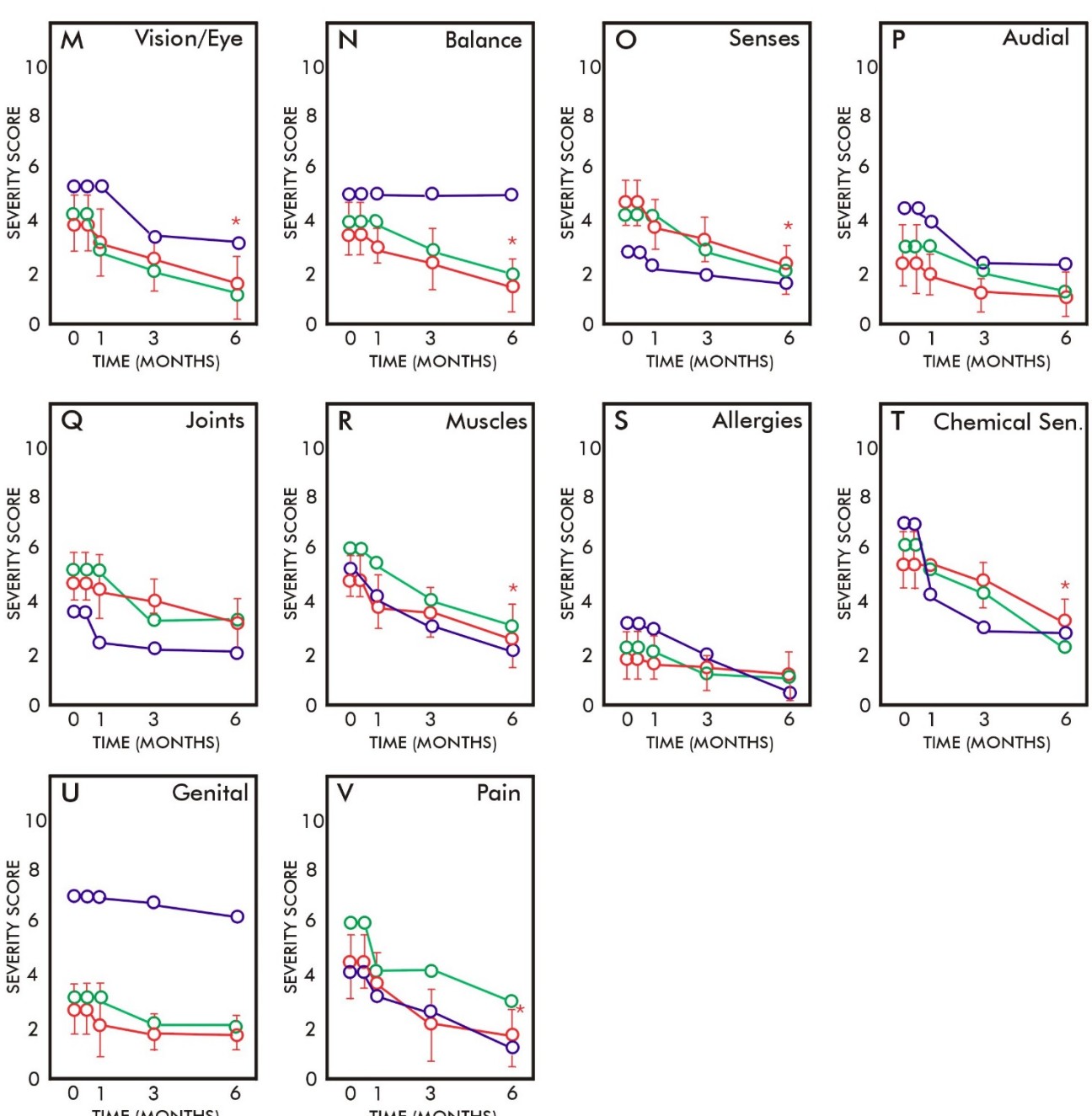

**Figure 3.** Symptom category scores with time reported by Study Participants who took the oral MLR supplement NTFactor Lipids® (6 g per day). The mean symptom category severity scores (±standard deviations) of all trial participants (red symbols) are compared to two individual participants (green and blue symbols) before the trial, and at one week, one month, 3 months and 6 months. (**A**) Heart symptoms, (**B**) nasopharyngeal symptoms, (**C**) breathing difficulties, (**D**) neurologic symptoms, (**E**) sleep disturbances, (**F**) fatigue, (**G**) infection(s), (**H**) hair/scalp disturbances, (**I**) skin disturbances, (**J**) gastrointestinal symptoms, (**K**) urinary symptoms, (**L**) oral cavity disturbances, (**M**) vision/eye disturbances, (**N**) balance disturbances, (**O**) sense disturbances, (**P**) audial disturbances, (**Q**) joint symptoms, (**R**) muscle symptoms, (**S**) allergies, (**T**) chemical sensitivities, (**U**) genital disturbances and (**V**) pain. (*, $p < 0.01$; **, $p < 0.001$).

## 4. Discussion

Gulf War veterans were exposed to a number chemical, biological, radiological, physical and mental stressors during the conflict [1,2,7–13]. Thus, GWI patients may have been exposed to a variety of multiple agents and stressors, and this makes any approach to treatment much more difficult. Although there is apparently no single cause of GWI, there are treatment approaches that can reduce the severity of signs and symptoms of this chronic condition [7,13]. These approaches are often based on documented or presumed exposures during the conflict, such as exposure to specific infections, certain chemicals and neurotoxicants, physical trauma, stress and depleted uranium, among other exposures [7,9,12,13,23–28].

Other approaches to treatment of GWI have been based on loss of function. For example, Koslik et al. [29] found that mitochondrial function in GWI patients was impaired by examining post-exercise phosphocreatine recovery time. Since loss of coenzyme Q10 (CoQ10) is often associated with loss of mitochondrial function [30], this information was used to design a clinical trial using supplementation with CoQ10 [31]. Moreover, there are several different natural supplements that have been used to treat mitochondrial dysfunction [32,33]. One of the most effective natural supplements to treat mitochondrial dysfunction is a mixture of membrane glycerolphospholipids (NTFactor Lipids®) which repairs the inner membrane and restores mitochondrial inner membrane trans-membrane potential necessary for mitochondrial function [34].

Thus, use of the natural MLR glycerolphospholipid mitochondrial supplement in Gulf War veterans with GWI was first examined in case studies [18,20]. Various dose levels were tried over time in order to determine a daily dose level that reduced various symptoms. Although dose levels lower than 6 g per day have been successfully used to reduce symptoms such as fatigue in patients with chronic fatigue [34], other symptoms, such as widespread pain found in fibromyalgia patients, require a higher daily dose of glycerolphospholipids to be effective [18,20]. Here, we found that 6 g per day of the test supplement reduced pain in GWI patients, but it took 3 months for the reduction of pain severity to reach significance. This could be due to a minor subfraction of NTFactor Lipids® that is likely to be involved in pain reduction by its interaction with plasma membrane channels such as the TRP membrane channels in peripheral nerve cells [16].

The present study, although important as a preliminary clinical study, had a number of limitations. This was not a controlled clinical trial, and there were limited numbers of participants that completed the study. The trial only examined one dose level of the MLR glycerolphospholipid supplement, which we determined in case studies to be sufficient to reduce pain and other symptoms in Gulf War veterans. Additionally, the study could have benefited from the addition of other mitochondrial natural supplements, as we have done in other studies, including those with NTFactor Lipids®, CoQ10 and other mitochondrial supplements [35]. Here, we found that MLR with oral glycerolphospholipids appeared to be a simple, safe and potentially effective method of slowly reducing the severities of multiple symptoms in chemically exposed veterans. There were significant reductions in symptom category severities related to fatigue, pain, breathing, vision, sleep, chemical sensitivities and other symptoms during the study. Most if not all of these symptoms are also related to mitochondrial function, and MLR supplements like the test supplement are known to enhance mitochondrial function [32–34]. There are a number of other possible mitochondrial and cellular membrane supplements that could be useful in various combinations for reducing symptoms in GWI patients, and these should also be investigated [35,36].

**Supplementary Materials:** The following supporting information can be downloaded at: https://www.mdpi.com/article/10.3390/ijtm2020014/s1.

**Author Contributions:** Both authors (G.L.N. and P.C.B.) contributed to planning, execution, and evaluation of the study and drafting and revising the manuscript. All authors have read and agreed to the published version of the manuscript.

**Funding:** G.L.N. acknowledges support from the Institute for Molecular Medicine, and the study test product was provided by Nutritional Therapeutics, Inc.

**Institutional Review Board Statement:** This study was IRB approved.

**Informed Consent Statement:** This study included informed consent, and each participant was required to sign an informed consent document.

**Data Availability Statement:** For further information contact the corresponding author.

**Acknowledgments:** The authors thank J. Michaels and R. Fielding for assistance.

**Conflicts of Interest:** G.L.N. is a part-time consultant to Nutritional Therapeutics, Inc. No other possible conflict of interest are reported.

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
