# Peer review of "Membrane Lipid Replacement with Glycerolphospholipids Slowly Reduces Self-Reported Symptom Severities in Chemically Exposed Gulf War Veterans"

_2673-8937, doi:10.3390/ijtm2020014_

Round 1
Reviewer 1 Report
This is an important clinical study that tests the efficacy of an oral food supplement on clinical symptoms in a small group of subjects with common histories stemming from a combat situation some 30 years ago.
The methods section should be modified to include more information as to the patient symptom survey form (I was unable to access Supplemental Fig.1) Details are needed on the time points over which a subject completed a survey. i.e it is not clear if, when a subject is recording the severity of symptoms, these symptoms were experienced at the time of survey or in the 24 hours up to the time of the survey or in the previous seven days before the time of the survey, for example.
The Discussion fails to state the authors' conclusions from the results. The Abstract gives a brief summary of the authors' conclusion in lines 19-22, but no similar statement is found in the Discussion, nor is there a dissection of the statistical strength of the findings reported in the Results. This oversight should be rectified
Author Response
REVIEWER 1
This is an important clinical study that tests the efficacy of an oral food supplement on clinical symptoms in a small group of subjects with common histories stemming from a combat situation some 30 years ago.
The methods section should be modified to include more information as to the patient symptom survey form (I was unable to access Supplemental Fig.1) Details are needed on the time points over which a subject completed a survey, i.e it is not clear if, when a subject is recording the severity of symptoms, these symptoms were experienced at the time of survey or in the 24 hours up to the time of the survey or in the previous seven days before the time of the survey, for example.
Response: The Supplemental Fig. 1 was submitted at the time the manuscript was submitted. The instructions in the survey form as well as additions in Section 2.2 should clarify the times when the survey form was filled out. The respondents were told to report their signs and symptoms severities as indicated in the survey form within 24 h of the appropriate dates. Due to the length of time over the course of this study (6 months), it is extremely unlikely that this range of time (24 h) had any effect on the results.
The Discussion fails to state the authors' conclusions from the results. The Abstract gives a brief summary of the authors' conclusion in lines 19-22, but no similar statement is found in the Discussion, nor is there a dissection of the statistical strength of the findings reported in the Results. This oversight should be rectified.
Response: Conclusions were added to the Discussion. A sentence or two was also added to the Discussion summarizing the over-all results, their significance and links to mitochondrial function.
Reviewer 2 Report
First off, I would like to congratulate the authors on their work and the aim of the study, to search for possible treatments for veterans affected by the Gulf War Illness. I acknowledge the author's expertise and experience on the topics of the study, both Gulf War Illness and cell membrane biochemistry. The study provides meaningful evidence to support a potential alternative therapy for Gulf War Illness-affected individuals and is of relevance to the journal. The manuscript is presented with rigor, clarity and generally well-structured. Unfortunately, some changes are needed to improve its clarity and quality. Please follow the suggestions provided below:
INTRODUCTION
Although it is a controversial topic, perhaps it would be best to reduce information on military-related information, regarding the Kuwaiti Theater of Operations, the name of the military operation, or the terms of “military support”. Perhaps the most relevant aspect regarding the ailments’ nature of the affected veterans would be a brief description of the involved chemical and/or biological agents to which these individuals were exposed (e.g. sarin gas). In addition, could the authors elaborate a bit further on Membrane Lipid Replacement as a potential therapy? References provided by the authors are excellent sources of information on this, but they could aid the reader to have a better perspective on the topic and methods. Information on the supplement used in the study (types of phospholipids, natural origin, etc) should perhaps be included here, instead of in Materials and Methods.
Besides these minor issues, the Introduction provides background and context to the topic in an excellent manner.
MATERIALS AND METHODS
Information on materials and methods appears well-structured and with enough detail and clarity. I may only suggest if the authors could elaborate, even if briefly, on the statistical model used and state the software used.
RESULTS
L148: Given that abbreviations have been introduced above in the text, could the authors use GWI for Gulf War Illness?
Figure 1: In order to better interpret the data presented, would the authors be so kind as to present this information as a table?
Figure 2: Same comments as for Figure 1. Could the authors express this data as a table?
Figure 3: It is, unfortunately, unclear to what group the individuals represented in green and blue belong. Were these two individuals part of an outside group, since it is a non-controlled study? Please check and clarify in the text.
L221-234: Considering the links of one of the authors with the provider of the supplements and how this information is presented, it seems to be a bit biased towards a specific commercial product (NTFactor Lipids), which appears directly mentioned as much as 7 times in this section. Perhaps it would be better to establish a discussion of the results in relation to the proper glycerophospholipids and/or in combination with fructo-oligosaccharides as a formulation, rather than discussing a specific commercial product itself. In addition, a mention of phospholipids intake through diet would be desirable.
REFERENCES
References are generally in order and follow the journal’s guidelines. Yet, could the authors include DOIs to each reference, when available. This could help access the cited works more easily. In addition, please check and remove some links appearing in L296, 308, 328 and 340.
FINAL REMARKS
The work is of interest to the journal and of good quality. Some improvements are needed to improve its structure and clarity prior to acceptance and publication. Therefore I suggest something in between major and minor revisions.
Author Response
REVIEWER 2
First off, I would like to congratulate the authors on their work and the aim of the study, to search for possible treatments for veterans affected by the Gulf War Illness. I acknowledge the author's expertise and experience on the topics of the study, both Gulf War Illness and cell membrane biochemistry. The study provides meaningful evidence to support a potential alternative therapy for Gulf War Illness-affected individuals and is of relevance to the journal. The manuscript is presented with rigor, clarity and generally well-structured. Unfortunately, some changes are needed to improve its clarity and quality. Please follow the suggestions provided below.
Response: We thank the reviewer for his/her kind comments. Although this is a preliminary study, we felt that the results were important enough to report and hopefully generate enough interest and support for more elaborate future studies.
INTRODUCTION
Although it is a controversial topic, perhaps it would be best to reduce information on military-related information, regarding the Kuwaiti Theater of Operations, the name of the military operation, or the terms of “military support”. Perhaps the most relevant aspect regarding the ailments’ nature of the affected veterans would be a brief description of the involved chemical and/or biological agents to which these individuals were exposed (e.g. sarin gas). In addition, could the authors elaborate a bit further on Membrane Lipid Replacement as a potential therapy? References provided by the authors are excellent sources of information on this, but they could aid the reader to have a better perspective on the topic and methods. Information on the supplement used in the study (types of phospholipids, natural origin, etc) should perhaps be included here, instead of in Materials and Methods.
Besides these minor issues, the Introduction provides background and context to the topic in an excellent manner.
Response: We have added additional information in the Introduction. Since there were only a couple of lines in the introduction related to the specific military operation and its location, we decided to leave these in the manuscript. The reason for this is that time and place of environmental exposures are important. The Kuwaiti Theater of Operations (KTO) defines specifically the location of exposures (Kuwait, Southern Iraq and the Far North of Saudi Arabia) and the Operation defines the date (Jan-Feb 1991). Note that other operations (Operation Desert Shield, Operation Badger, etc.) occurred at different times in the same region. We also feel that the definition of Membrane Lipid Replacement (MLR) is in the Introduction, along with the reference [18] of a case series where MLR with the same supplement was used to reduce symptom severities in Gulf War veterans. The case presentation in Ref. 18 suggested that the higher dose level of MLR used in the present study was more appropriate than the dose level used in other conditions to reduce symptom severities (reviewed in Refs. 14-16).
MATERIALS AND METHODS
Information on materials and methods appears well-structured and with enough detail and clarity. I may only suggest if the authors could elaborate, even if briefly, on the statistical model used and state the software used.
Response: We have added additional information on the statistics and the program used in the Methods.
RESULTS
L148: Given that abbreviations have been introduced above in the text, could the authors use GWI for Gulf War Illness? OK
Response: OK, done.
Figure 1: In order to better interpret the data presented, would the authors be so kind as to present this information as a table?
Figure 2: Same comments as for Figure 1. Could the authors express this data as a table?
Response: After carefully considering the Reviewer’s request to change Figs. 1 and 2 into tables, we decided not to do this. The data are presented clearly, and we strongly doubt that the data would be any easier to see and digest as two tables. Also, presenting the data in tables would increase the manuscript length.
Figure 3: It is, unfortunately, unclear to what group the individuals represented in green and blue belong. Were these two individuals part of an outside group, since it is a non-controlled study? Please check and clarify in the text.
Response: We have reworded the Results section to clarify that the two individual patient examples in Fig. 3 were members of the study group. We felt that readers would appreciate that there were individual differences in different veterans. Of course, the important data are the group data (shown as RED symbols).
L221-234: Considering the links of one of the authors with the provider of the supplements and how this information is presented, it seems to be a bit biased towards a specific commercial product (NTFactor Lipids), which appears directly mentioned as much as 7 times in this section. Perhaps it would be better to establish a discussion of the results in relation to the proper glycerophospholipids and/or in combination with fructo-oligosaccharides as a formulation, rather than discussing a specific commercial product itself. In addition, a mention of phospholipids intake through diet would be desirable.
Response: We have reduced by 6 the number of times that the specific test supplement NTFactor Lipids was mentioned in the Methods, Results and Discussion. It is almost impossible to ingest daily enough of some of the specific glycerolphospholipids found in NTFactor Lipids using diet alone. This is discussed in the references [14-16]. For example, it would require eating over 15 Kg of soybeans daily to obtain a dose of 1.8 g of membrane glycerolphospholipids, and most of these would be phosphatidylcholine, with very little phosphatidylglycerol, an important precursor of mitochondrial cardiolipin. The glycerolphospholipid composition of NTFactor Lipids is balanced to be similar to the composition of human mitochondrial inner membrane phospholipids.
REFERENCES
References are generally in order and follow the journal’s guidelines. Yet, could the authors include DOIs to each reference, when available. This could help access the cited works more easily. In addition, please check and remove some links appearing in L296, 308, 328 and 340.
Response: We have added doi (or in some cases URLs) when available for the copy editor so that they can be converted to cross ref or other tools.
FINAL REMARKS
The work is of interest to the journal and of good quality. Some improvements are needed to improve its structure and clarity prior to acceptance and publication. Therefore I suggest something in between major and minor revisions.
Reviewer 3 Report
The article is written in a clear and understandable way. It is a very valuable preliminary report that needs to be followed up with studies with a control group. The report has some limitations (especially the control group, which does not eliminate the placebo effect), which are correctly pointed out in the last paragraph of the article.
Author Response
REVIEWER 3
The article is written in a clear and understandable way. It is a very valuable preliminary report that needs to be followed up with studies with a control group. The report has some limitations (especially the control group, which does not eliminate the placebo effect), which are correctly pointed out in the last paragraph of the article.
Response: We thank the reviewer for his/her comments. We agree with the comments provided by the reviewer.
We thank the Reviewers for their constructive comments. We feel our manuscript is now ready for publication.